# IoT-Based Systems for Soil Nutrients Assessment in Horticulture

**DOI:** 10.3390/s23010403

**Published:** 2022-12-30

**Authors:** Stefan Postolache, Pedro Sebastião, Vitor Viegas, Octavian Postolache, Francisco Cercas

**Affiliations:** 1Instituto de Telecomunicações, ISCTE-Instituto Universitário de Lisboa, 1649-026 Lisboa, Portugal; 2Instituto de Telecomunicações, Portuguese Naval Academy, 2810-001 Almada, Portugal

**Keywords:** NPK sensor, smart sensors, Internet of Things, mobile application, horticulture, soil nutrient assessment, precision agriculture

## Abstract

Soil nutrients assessment has great importance in horticulture. Implementation of an information system for horticulture faces many challenges: (i) great spatial variability within farms (e.g., hilly topography); (ii) different soil properties (e.g., different water holding capacity, different content in sand, sit, clay, and soil organic matter, different pH, and different permeability) for different cultivated plants; (iii) different soil nutrient uptake by different cultivated plants; (iv) small size of monoculture; and (v) great variety of farm components, agroecological zone, and socio-economic factors. Advances in information and communication technologies enable creation of low cost, efficient information systems that would improve resources management and increase productivity and sustainability of horticultural farms. We present an information system based on different sensing capability, Internet of Things, and mobile application for horticultural farms. An overview on different techniques and technologies for soil fertility evaluation is also presented. The results obtained in a botanical garden that simulates the diversity of environment and plant diversity of a horticultural farm are discussed considering the challenges identified in the literature and field research. The study provides a theoretical basis and technical support for the development of technologies that enable horticultural farmers to improve resources management.

## 1. Introduction

Accelerated progress over the past 50 years produced by new technologies in industry, healthcare, or tourism have paved the way for transforming the way of producing our food. The European Farm to Fork Strategy [1] aims to promote scientific discoveries and implementation of new technologies (i.e., new machinery, genetically improved plants or animals, and information and communication technologies for efficient and sustainable farming), and to increase awareness and demand for sustainable food. Many policy makers and farmers are currently aware of potential benefits of introducing new technologies that may overcome challenges of contemporary agriculture (i.e., technology for efficient and sustainable use of resources; technology for better risks and variability management that optimize yields and improve economics). For instance, it is well known nowadays that extensive and excessive use of synthetic fertilizers, irrigation practices in intensive agriculture, in addition to consequences derived from climate changing are associated, in many regions of the world, with soil and water depletion [2] and, subsequently, great reduction in crop yields. By implementing technologies that optimize the control of irrigation systems, it is possible to: (i) grow more crops; (ii) produce higher quality crops; (iii) have “insurance” against seasonal variability and drought; (iv) maximise benefits of fertilizer applications; and (v) use areas that would otherwise be less productive, among others [3].

Horticulture is a branch of agriculture that includes pomiculture (cultivation of fruit trees), olericulture (cultivation of vegetables), viticulture (cultivation of grapevines), floriculture (growing flowers, some with medicinal use), and gardening (the art and craft of laying out and care of a plot of ground devoted partially or wholly for growing plants, such as flowers, herbs, or vegetables) [4]. Horticultural crops cover vegetables, fruits, mushrooms, and condiments or medicinal plants. The importance of horticultural crops is currently underscored by progress in nutrition science. It is well understood nowadays that longevity of the healthy people, as well as prevention and treatment of diverse human pathology, is greatly associated with consumption of a variety of vegetables or fruits. The concept of “healthy eating plate” [5] underlines the importance of vegetables and fruits for healthy balanced meals. However, the increased need for global commercially available vegetables and fruits (associated with change in pattern of human food consumption, as well as the growing global population) has been raising new challenges in horticulture practices. For instance, the use of water, fertilizers, and pesticide in intensive horticulture may have a great impact on ground water [6] or may produce soil pollution [7]. Therefore, research is conducted to find solutions for better management of water, fertilizers, and pesticide in horticulture. As such, determinants for soil nutrients mining or depletion (defined as removal of more soil nutrients by crops than added through manure or fertilizers) [8] are investigated. Furthermore, many studies were published related to water use and technology that may optimize irrigation for diverse crops [9,10,11]. The first step that is necessary to implement an efficient water and fertilizer management system is field information collection. Various techniques and technologies have been developed in the past decades to collect data on crop survival environments and growth condition. Interest is growing on technologies based on Internet of Things (IoT) that provide means for researchers and farmers to find solutions for better management of crop inputs (e.g., reduce quantity and rate of fertilizers without sacrificing food production). In the last years, various studies have described different IoT-based systems for agriculture. Advances in sensors technologies, connectivity and data storage, data analytics, and algorithms for decision making and prediction enable nowadays the creation of intelligent management systems for water use or irrigation, or for optimal fertilizers or pesticide dispersion. However, no large-scale or commercially available systems are currently in use for many horticultural crops. Why does this happen? We investigate and we describe in this work our finding related to the challenges of implementing an IoT-based system in horticulture. The main focus of our research is on a soil nutrients management system, as this topic has scarce information, despite its importance for crop productivity.

In the following sections, we present: an overview of different techniques for soil fertility characterization (Section 2); a discussion on different challenges for implementation of IoT-based systems for soil nutrient assessment (Section 3); a description of the architecture of our developed IoT system for soil nutrient characterization (Section 4); a presentation of some results obtained by our system that underscore the challenges related to the IoT-based development (Section 5); a discussion on the results and potential solutions that can increase the effectiveness of the IoT-based system for soil nutrients assessment (Section 6); and, finally, our conclusions.

## 2. Techniques and Technologies for Soil Fertility Characterization

The Food and Agriculture Organization (FAO) defines soil fertility as “the capability of soil to sustain growth by providing essential plant nutrients” [12], and, throughout the 17th and 18th century, it was believed that this was derived by one substance, the “humus”. It was not until the 19th century that we started to unveil the relationship between soil nutrients and fertility. The maintenance of soil fertility in agricultural systems involves the use of manure and other organic materials, inorganic fertilizers, lime, the incorporation of legumes into the cropping systems, or a combination of these. For optimal production and to preserve the fertility of the soil, application of fertilizers to maintain adequate amounts of all necessary nutrients should consider two principles. The first one proposed by Justus von Liebig, the law of minimum, states that “the growth of the plant is limited by the plant-nutrient element present in the smallest quantity, all other being present in adequate amounts” [13]. Excessive fertilization may have a great impact on soil properties and decrease growth. The second principle, the Mitscherlich law of diminishing yield increment, states that, by increasing the quantity of a given nutrient, the increment in crop growth will decrease until a point where the maximum yield capacity has been reached, which can be described by the following equation [14]:(1)$$y=A\;\left(1-e^{-Cx_{t}}\right),$$
where *y* is the yield, *x_t_* is the total nutrient quantity, which includes the plant available quantity of nutrient in the soil and the added quantity of a nutrient, *A* is the maximum attainable yield, given that *x_t_* is present in adequate quantity, and *C* is the efficiency factor. When the total amount of a nutrient in the soil—both naturally occurring and added—exceeds the needs of the crop, it starts to limit yield, as it either becomes toxic or inhibits the absorption of other essential nutrients. In this case, the crop response or the relation between added nutrients and plant growth can be described using polynomial equations of type [14]:(2)$$y=a_{1}+b_{1}x+c_{1}x^{2}$$
where *y* is the yield, *a*_1_ is the estimated plant available quantity of a nutrient in the soil, *x* is the added nutrient quantity, and the maximum yield (*A* from Equation (1)) is given by the value of –(*b*_1_/2*c*_1_). Figure 1 represents the crop response when applying equations 1 and 2.

This method has some shortfalls, namely it does not take into account spatial variability and the influence that external factors, such as water stress, pests, and diseases, have on the nutrient uptake and availability, and thus on plant growth [15]. However, it is still an important tool for characterizing soil fertility and it is the standard used to benchmark other soil fertility evaluation systems.

Soil fertility characterization may be performed through (i) qualitative techniques (e.g., observation of soil colour, hand test for soil texture characterization, observation of abundance, and diversity of soil flora and fauna); (ii) quantitative techniques (e.g., quantification of macro- and microelements from the soil and quantification of soil water content); and (iii) semi-quantitative technique (e.g., questionnaire or data evaluation on soil properties, rate, and quantity of fertilizer application).

Techniques for soil fertility characterization could be grouped also on: (i) biological characterization; (ii) plant analysis; and (iii) chemical and physical soil parameter evaluation. For biological characterization of soil fertility, the evaluation of changes in abundance and diversity of soil fauna and flora, microbial biomass, and enzyme activity give important information on the health of soil, on the depletion of soil nutrients, or impact of fertilizers on soil. Soil fauna and microbial communities play a key part in litter decomposition to inorganic forms that can be absorbed by the growing plants. In addition, the activities of soil fauna improve soil water infiltration and storage [16]. Qualitative methods (e.g., observation of soil sample) are used for biological characterization of soil fertility that requires evaluator expertise in soil evaluation. Evaluator fatigue and lower capacity for visualization of sample (e.g., lower luminosity in the environment) may introduce error in the evaluation. Furthermore, qualitative and semi-quantitative methods may be used on plant analysis. Observation of plants can be used to determine critical nutrient conditions by the observation of visible symptoms, usually associated with nutrient deficiency. Figure 2 shows a healthy tobacco sprout and the characteristic decolorated tobacco leaf when nitrogen deficit in soil exists.

Recently, plant analysis for soil fertility characterization may be conducted using satellite imagining (e.g., using MODIS—moderate-resolution imaging spectral radiometer information on normalized difference vegetation index and some soil properties) [18] or using drone—unmanned aerial vehicle (UAV) with attached camera (i.e., high-resolution RGB camera, hyperspectral camera, or thermal camera) [19,20]. Resolution of satellite images is measured in meters, while drone image resolution may be in the order of centimeters. A limitation of this plant analysis method is that, when indicators of nutrient deficiency are visible, it is already too late to correct the decline in fertility. In addition, a quick testing of the quantity of a certain nutrient in the plant can be included in the analysis. However, plant analysis confers only a semi-quantitative dimension on soil fertility because the obtained data depend on the ratio of nutrients inside the plant, and nutrient quantity on a plant varies with species, time of sampling, physiological maturity, plant part sampled, and incidence of diseases.

Chemical and physical soil parameter evaluation can be conducted using various techniques and technologies. Soil characterization is typically accomplished through the analysis of a soil extract, from 20 to 50 samples collected from squared areas of 20 acres (approximately 1 ha) from the field at different depths (0 to 20 cm). Analysis is performed by combining water, chemical extractants, ion-exchange resin and membranes, and electro-ultrafiltration through various laboratory methods, such as colorimetry and spectroscopy [21], which are very time-consuming and expensive. Moreover, in the majority of cases, soil tests are performed on a gravimetric basis, i.e., considering the soil’s mass, instead of volumetric, considering the soil’s volume. Soil volume may vary with the bulk density and depth of sampling. Moreover, the assessment should consider the great differences in soil chemical and physical properties on samples collected from the surface in comparison with samples collected from deeper layers. Moreover, in soil sample collection, the type of roots of plant on culture should be considered (i.e., for plants with taproots, soil sample should be collected from a deeper layer than for plants with fibrous roots). Sample depth must remain consistent because many soils are stratified, and this may introduce more variability in measurements and inadequate interpretation of data. Furthermore, based on the results, corrective measures may be applied to the whole area from which the samples were collected in a uniform manner.

Advances in portable sensors and spectral sensing techniques enable soil assessment on site, estimation of nutrients on the go, and analysis of flow on soil nutrients for a specific site. Moreover, geographic information systems (GIS) can be used to map in real time the measured values of nutrients to specific locations or areas, which allows better management of fertilizer application.

Table 1 summarizes the advantages and disadvantages of each fertility assessment method.

In the next subsections, nutrients and factors that reduce plant uptake are explored, as well as sensor types and/or systems that are typically used to monitor its magnitude.

Soil Nutrients

Soil is a complex structure which includes various components in solid, liquid. And gas form. Out of all the nutrients present in the soil, 19 nutrients are considered to be important for plant development, which can be categorised into essential elements and beneficial elements (Table 2). Essential elements are irreplaceable elements involved directly in the metabolism of the plant, without which plants cannot complete their lifecycle. Essential organic elements comprise carbon (C), hydrogen (H), and oxygen (O). These elements are mostly absorbed by the plant from the atmosphere through its leaves. However, a small quantity of these nutrients can also be found in the soil as part of the soil organic matter (SOM) in the case of C or in the form of water in the case of H and O, which cannot be directly consumed by the plant through its roots.

Inorganic/mineral nutrients are absorbed through the roots of the plant and are important constituents of plant tissue. However, some nutrients are used in greater quantities, as they are essential parts of plant organic compounds(light-green), such as proteins and nucleic acids, or they play an important role in regulating the pH and osmotic potential of the cells, called macronutrients [23]. Macronutrients (orange cells) can be (i) primary—nitrogen (N), phosphorous (P), and potassium (K), who’s absence often limits plant growth; or (ii) secondary—calcium (Ca), magnesium (Mg), and S, that rarely limit plant growth. Micronutrients (yellow-brown), on the other hand, are required in significantly lower quantities (boron (B), chlorine (Cl), copper (Cu), iron (Fe), manganese (Mn), molybdenum (Mo), nickel (Ni), and zinc (Zn)), as they are components of enzymes (Table 2). The average concentrations that can be found in plants of these elements (Table 3) is a good indicator of their importance for plants as well and gives us an idea of the ratio of nutrients that should be available in the soil. Aside from essential elements, there are also beneficial elements (light-blue) (sodium (Na), cobalt (Cb), silicon (Si), and aluminum (Al)), since only a small number of plant species have been found to require them [24].

Even if nutrients are available in certain quantities in the soil, not all can be consumed by the plant. Nutrient uptake depends on properties of the plant, such as species and maturity [14], and external factors, such as soil acidity, salinity, temperature [25], moisture, and concentration of other ions. For instance, ions with similar physicochemical properties, such as K^+^ and Na^+^, will compete for entering the root. In addition, factors such as oxygenation, air temperature, as well as pests and diseases may impair metabolic processes and affect the root system. Components inside the plant, such as carbohydrates (sources of energy) and oxygen, are necessary to absorb ions from the soil [15,26]. For this reason, in order to define the corrective measures (fertilizer quantity) that need to be applied to a certain site, it is not enough to determine the current soil nutrient content. Other factors need to be related to the historical nutrient data in order to identify patterns in nutrient uptake [27].

## 3. Information System for Soil Nutrients Assessment

In recent years, the increasing awareness of the impact that intensive agricultural practices have had on the environment and on our health, as well as the pressure placed on farmers to increase their production to meet the needs of an ever-growing population have led to the necessity to control the agricultural production processes more thoroughly. Recently, various studies have emphasized the importance of information and communication technologies to build information systems for monitoring crops, which support farmers for better management of resources, such as optimal use of water and fertilizer in different crops at different states of growth and environmental conditions. Information system protocols comprise different layers: (i) objects/access layer/edge technology; (ii) network layer; (iii) co-ordination layer/service management; and (iv) application layer.

An important component of the information system is the structure that allows data collection. Information on soil nutrients could be obtained by direct measurement using specific sensors or by indirect measurements using data from different sensors and aggregate data from different sources (e.g., using public weather, data introduced in the system by the farmer on date, and quantity and type of administered fertilizer). Methods for monitoring soil fertility can be grouped into three categories: (i) in situ, (ii) proximity, and (iii) remote sensing.

In situ sensing uses sensors that are nondestructive and easy to use in different terrains and different environmental conditions. *Point-based* measurements are obtained using sensors that are in close contact or at a small distance from sample. These techniques are used to measure soil parameters, such as moisture and nutrient content, pH, salinity, and temperature. Several techniques for measuring soil moisture are already used in agriculture, such as time-domain reflectometry (TDR), heat pulse probe, and tensiometric techniques, and many others are at early stages of development, such as capacitance, frequency-domain (FDR) reflectometry, neutron scattering, electrical resistivity tomography, and near-infrared reflectance technologies [28]. Sensing technology for soil macronutrients, nitrogen/nitrate (N), phosphorous (P), and potassium (K), also known as NPK content, is in an early stage. A search on IEEE Xplore Digital Library in 2022 using keywords “NPK sensor” retrieved 49 papers, the majority (48) presented in conference. In 24 papers from 49, technology is presented that includes NPK sensor. The majority of the works (15 from 24) are proof-of-concept papers that do not give information on calibration or results on sensitivity or accuracy. Several papers (11) present their results on nutrient sensing using laboratory experimentation. Data on field measurements using NPK sensor are very scarce (one presents collected data for 1 month in a vineyard but does not give information on type of NPK sensor [29]; one presents data on six soil samples from no identified field but is not clear if measurements were performed in the laboratory or in the field [30]; one presents data from nine soil samples from parks and alongside the street, also with no clarity on the place where measurements were performed [31]). There are different technologies for soil N, P, and K quantification, such as ion-selective membrane (ISM)-based electrochemical sensors, enzyme-based biosensors, molecular-imprinted polymers (MIP)-based biosensing approaches, electro-reduction of nitrate to ammonium ion on copper-based electrodes [32], fluorescence-based sensors, tamer-based sensors, on-the-go spectroscopy, and electrophoresis-based methods [22]. Optical or colorimetric sensor [31,33,34,35,36,37,38,39,40,41,42,43], conductivity sensors [30,44,45], and electrochemical sensor [46,47,48,49] were used for N, P, and K quantification as the sensing layer of an information system based on Internet of Things technology. The proposed sensing devices for IoT-based information systems for soil nutrients assessment have several limitations. Various proposed optical and colorimetric sensing devices enable measurements only after soil sample preparation (i.e., dilution, filtration, and addition of different reagents to the sample that enable optical detection of the targeted soil element). The procedure is similar as in traditional laboratory measurements and requires laboratory conditions, employing expensive materials and trained and experienced technicians. Moreover, the proposed devices did not consider the progress in spectrophotometry technology (i.e., the innovation in this technology increased accuracy of measurement but also improved the feasibility and maintenance of the equipment). Proposed miniaturization in recent studies that include optical or colorimetric sensors in IoT-based systems is made at the expense of the feasibility of the equipment. Furthermore, direct measurements are proposed in some work using an optical sensor by putting the sensor at a small distance to the soil. A number of variables may influence those measurements (e.g., environmental light; soil texture; and chemical composition) and, therefore, the inaccuracy of the measurements is expected to be higher than in laboratory setting. For easy quantification of a certain substance, different electrochemical sensors were developed in recent years. Many of these sensors have high sensitivity and good selectivity. However, at least for some years in the future, the price of electrochemical sensors will be high, as the production of these sensors requires expensive equipment, expensive materials, and high professional expertise. Moreover, their durability is generally low, and many electrochemical sensors would not withstand long-term burial in soil for in situ measurements. The main advantages of conductive sensors for soil nutrients measurement are durability, low cost, and easiness of use. However, erroneous measurements in substrates with complex composition are a well-known limitation of this type of sensor. Soil parameters, such as pH, salinity, permeability, moisture, and chemical composition, may affect the estimation of soil N, P, and K using conductive sensors. Our hypothesis is that a low-cost measurement system using an NPK conductive sensor could be developed if data from the sensor would be aggregated with data from environmental conditions to enhance the accuracy of N, P, and K quantification. As such, in the present work, we propose a multi-channel sensing device for N, P, and K estimation.

Data on soil pH are important both because different plants required different soil acidity but, also, because pH influences nutrient availability for the plants. Current techniques used for the measurement of soil pH are: (i) electro-chemical, such as glass electrode, organic modified electrodes (OCPMEs), ion-sensitive field effect transistor (ISFET) and pH image sensor; (ii) optical, such as optical fiber sensors, fluorometric sensors, holographic sensors, ratiometric pH-dot sensors, and camera sensors; (iii) conductimetric; (iv) potentiometric; and acoustic (v), such as cantilever and microcantilever sensors and magnetoelastic pH sensors [50].

In situ observations of soil water and nutrients content can provide more accurate data than remote sensing (satellite data acquisition), and these can even serve as reference calibrations and improve the accuracy of estimation of soil moisture using satellite data [51,52]. Although the data acquired by in situ sensors is richer and more accurate than satellite data (satellite radiometers responds to soil moisture only within the top 1 to 2 cm of the soil; image resolution measured in meters did not give comprehensive information for small size, hilly landform, and diverse plant culture), it is not easy to deploy a dense network of in situ stations for real-time ground-based soil water and nutrients content measurements. Since in situ sensors are used for point-based measurements and it is not viable to flood the field with sensor nodes, it is necessary to select strategic locations to place the sensors. To determine these locations, one approach would be to apply standard soil testing procedures to the soil, i.e., split the field into areas of up to 20 acres and collect from 15 to 20 cores of soil from each in a zigzag or random fashion, and select the locations from between the samples that present higher variability [53]. However, as this process is very costly and time-consuming, data from proximal or remote sensing that offer on-demand, large-scale data on the crop could be integrated in the information system for optimal positioning of in situ sensors or eventually replace these sensors [54].

Remote sensing methods are based on satellite technology and have application in many areas of soil monitoring. They are useful for regional and global measurements and are based on emitted or reflected electromagnetic (EM) energy from the soil surface. These methods can be roughly classified in two categories: (i) active methods where the reflected or scattered energy is recorded in response to incident energy and (ii) passive methods where sensors (such as radiometers) are used to detect the radiation emitted by the target, also known as the brightness or the brightness temperature of the target [55]. Techniques have been developed for observing soil moisture content (SMC) remotely in the following EM spectral ranges: visible, infrared/thermal, and microwave, where the soil moisture is determined based on the intensity variations of the radiation due to parameters such as dielectric constant, temperature, and thermal properties [56]. Secondary parameters, such as vegetation cover, surface roughness, and atmospheric effects, also play an important role in remote sensing successful soil moisture content retrieval [56]. Vast amounts of remote soil sensing data were obtained from various satellites [56,57]. Normalized Difference Vegetation Index (NDVI) images available from NASA’s Land [58,59], Atmosphere Near real-time Capability for EOS (LANCE) and data from Group on Earth Observation Global Agricultural Monitoring (GEOGLAM) [60] provide important data on green vegetation density, mapping the regions where plants/crops are thriving and where they are under stress (i.e., due to lack of water). Aside from soil moisture, satellite imagery has been used in determining other soil parameters, such as carbon content, salinity, pH, temperature [56,61,62], detection of pests, diseases, and pollinators [63,64], prediction of crop yield and natural disasters, and phenotyping of crops [65,66]. Although it may seem appealing to use these methods due to the scale of the measurements, they have course spatial and temporal resolution and are restricted to shallow penetration. Additionally, complex analysis techniques are required to extract value from the collected data. Moreover, due to its coarse spatial resolution, it is not fit for use in small horticultural land (e.g., park, botanical garden, small vegetable farms, etc.) as, usually, they are not large spaces and each species occupies a different area.

Proximity sensing of soil nutrient is characterized by not being in direct contact with the soil or crop it is monitoring and being in the proximity of it. These methods can be further sorted into stationary and mobile. Stationary methods stay in the same place while collecting data (e.g., weather station), while mobile methods are attached to a moving agricultural vehicle or unmanned aerial vehicle (UAV) while collecting data. These methods can be more flexible and convenient than in situ sensors, while offering high spatial and temporal resolution, and are not restricted to shallow penetration like remote sensing methods. In addition, proximity methods can be used in areas that are not covered by satellites. Therefore, proximal sensing methods are a promising solution for the future. In several research projects that developed electronic systems and software technologies for optimal irrigation, data on soil water content were combined with climatic data. Climatic/weather data are used in the calculation of the evapotranspiration of the crop. Evapotranspiration can be estimated using the eddy covariance system (CE) [67], Bowen Station [68], lysimeter [69], scintillometer [70], and digital image processing methods, such as leaf area index (LAI) [71,72], percentage of plant cover (PGC) [73], and plant effective diameter (PED) [74,75,76]. Weather stations are in fixed locations and offer very good results over large areas. For a horticultural farm, the measurement stations should be positioned on strategic locations to give relevant information.

Proximity mobile sensing was implemented by the attachment of sensors to UAVs or agricultural vehicles, such as tractors [77] or minirobot [35]. UAVs have the advantage of being able to access places that agricultural vehicles cannot reach, such as swamps, and being able to rapidly cover large swaps of land. Nevertheless, the use of UAV is regulated, and restrictions differ from country to country [78]. Most governments require that UAVs must be operated by certified operators [79,80]; this has a significant impact on the frequency, quality, and type of research that can be performed. Certain rules may apply only to a specific aircraft type, while others may apply to all aircraft operating in specified applications. These regulations may be limiting acceptance and adoption level of farmers and research applications.

There are several types of UAVs, which can be classified according to their design, degree of autonomy, size and weight, and power source [80]. UAVs can also be classified into fixed-wing, rotary-wing, and hybrids, such as vertical take-off and landing (VTOL) [81]. UAVs may have either fixed or rotary wings based on the aerodynamic features, which determines their flight characteristics and other parameters [82]. Rotary-wing UAVs are used more than fixed-wing drones. Although, fixed-wing drones cover more ground (battery lasts longer) and faster than a rotary drone, a rotary-wing drone has more degrees of freedom (it can fly in all directions) and is easier to deploy as it can perform vertical take-off; for this reason, it is more popular than fixed-wing [78]. Moreover, hybrids between fixed- and rotary-wing are emerging, such as the case of vertical take-off landing (VTOL) UAV. They combine the extended range, payload capacity, and speed of fixed-wing UAVs with the capability of vertical take-off of the rotary-wing UAV [83]. VTOL UAVs are a relatively new research topic and, for this reason, there are no solutions that use this type of UAV in measurements.

Digital image processing methods are used in combination with UAVs, which analyze images from RGB cameras, multispectral cameras, hyperspectral cameras, or from light detection and ranging (LiDAR) systems [84]. Most applications use RGB and multispectral cameras, and very little research uses hyperspectral or LiDAR cameras for their higher cost. However, the most predominantly used are RGB cameras due to their low cost, low weight, ease of use, and the simplicity of the image processing required. On the other hand, since images from these cameras are limited to the visible spectrum, they lack the resolution for phenotypical analysis or diagnosis of diseases, although there are some applications of RGB cameras in disease detection [85,86,87]. As such, RGB cameras are especially suited for the determination of canopy height and lodging, and for extrapolating the Digital Terrain Model (DTM) and Digital Surface Model (DSM) for the surveyed region. Multispectral cameras are sensitive to more radiation bands than RGB cameras, typically four or six, including the visible spectrum, and thus provide images with more resolution and have higher cost. Multispectral cameras can be broadband or narrowband, according to their bandwidth, with broadband multispectral cameras having more spectral range than narrowband [88]. Even so, for their added resolution, they are suited to applications related to crop phenotyping [89], detection of crop diseases [90,91,92,93], drought and stress detection, determination of growth vigor, estimation of nutrients, and yield prediction [94,95]. Hyperspectral cameras collect data in the form of a succession of 5 to 10 nm bands, resulting in a higher level of spectral and radiometric accuracy than multispectral cameras. These are suitable for phenotyping, pest and weed detection, disease detection, and estimation of nutrient status [96,97,98].

Another sensor that has been used in combination with UAVs is thermographic cameras. Thermographic cameras are used to assess the relative surface temperature of objects, such as the soil and water vapor. Thermal images from UAV have better spatial and temporal resolution than satellite images and, for this reason, are becoming a valuable source of information for agronomic applications [99]. Thermal imagery has been used in plant growth and vegetation water stress studies [100,101,102].

Although UAVs are very versatile and provide a high resolution in measurements, they are not suitable for some horticultural land (e.g., botanical garden and park) that cultivate many plant species with different heights, which makes it difficult to define a flight distance from the soil, which would fit the monitoring of all species. In addition, many horticultural entities are the home to rare bird species, which could damage the UAV and be harmed by the UAV, or the UAV could frighten the birds, which is not ideal, as they are an important part of the ecosystem.

## 4. IoT-Based System for Soil Nutrient Assessment

Considering literature research data on soil nutrient sensing, we developed an information system that enables in situ measurements of soil moisture, pH, and N, P, and K. We developed and IoT-based system that uses a wireless sensor network and data could be visualized on a mobile application. Aside for being able to measure moisture, pH, and N, P, and K concentration simultaneously, the sensors can also be used to measure soil conditions at multiple depths, which is useful for plants with deep roots. Additionally, air parameter measurement channels are considered, including relative humidity and temperature, to extract correlations between soil and air conditions and the plant stress level (e.g., water stress and macronutrient stress). The main goal of our research work is the development of an information system that may detect plant stress that may improve decision making on irrigation and fertilization in horticulture. The concept of stress in plants has evolved in the past decades. Stress of plant means “any unfavorable condition or substance that affects or blocs a plant’s metabolism, growth, or development” [103]. We focus our work on measuring abiotic stressors as heat, drought, flooding, plant inadequate pH, and soil nutrients. The environmental factors, when acting as a stressor for a plant, can cause cell disturbances. The plant response to different unfavorable conditions is related to an intricate network that combines cellular physiological and morphological defenses [104]. Different sensing devices were described as useful to characterize the stress in plants by quantifying stress conditions and stress-induced damage in plants. We present here the first prototype that we developed, having as the main objective assessment of soil nutrient and air conditions that may be used for future development of an information system that would include a decision-making algorithm that may consider stress of plant, the Justus von Liebig law of minimum, and the Mitscherlich law of diminishing yield increment, when water and fertilizers will be administered. The developed prototype for soil nutrients assessment was tested on Lisbon Tropical Botanical Garden, a site with many features that may be found in other horticultural places (i.e., different slope of terrain; different plant species cultivated in small land size; different soil chemical and physical properties required by different plant; and complex landscape).

The developed system includes a set of smart sensing nodes that contain soil and air sensing units. The smart sensing nodes perform data acquisition, primary data processing, and data communication using Wi-Fi/LoRa communication protocols. The general architecture of the implemented distributed smart system is presented in Figure 3.

The developed IoT system is characterized by heterogeneous communication protocols; however, for the current application, Wi-Fi was the chosen communication protocol, as the garden provides a set of hot spots which offer good Internet Wi-Fi coverage to mobile phones. Following, a general description of the system is presented, including the smart sensing nodes that connect to a Cloud Server used for storage and for data analysis and a mobile app for geographic visualization of data.

### 4.1. Smart Sensor Nodes

Smart sensor nodes have the responsibility in the system to aggregate data from multiple sensing channels and to send them to a processing unit, in this case, the Cloud Server. They comprise an ESP32 Wi-Fi/LoRa board, which is connected to a multielectrode sensor based on the TDR working principle, for simultaneous measurement of temperature, pH, and soil moisture content, conductivity, and N, P, and K content. The measured soil parameters and the metrological characteristics of the multichannel sensor (JXCT [72]) are presented in Table 4.

The seven-parameter sensor is characterized by RS485 communication protocol that provides a duplex communication in which multiple devices on the same bus can communicate in both directions. To connect the JXCT seven-parameter sensor to the embedded processing unit expressed by ESP32, an RS485 to UART converter based on MAX485 chip was used. Some characteristics of the used chip are a low-power and slew-rate-limited transceiver that works at a single +5 V power supply and the rated current is 300 μA. In the present case, the UART port was configured to 4800 bps. MAX485 transceiver draws supply current of between 120 μA and 500 μA under the unloaded or fully loaded conditions when the driver is disabled. Considering the JXCT sensor power supply of 12 V, a step-up voltage regulator 5 V to 12 V was considered as part of the smart sensor module that is connected to a 10 Ah power 5 V power bank. The power bank is charged using a portable solar panel associated to the smart sensor node. The 27.5 cm × 19 cm solar panel has four cells, which provide 13.2 W input power. The smart sensor module block diagram is presented in Figure 4, while the first implemented prototype is presented in Figure 5.

In Figure 5, the ESP32 can be observed with a display OLED 0.96″ that is exclusively used in debugging mode, considering that the smart sensing module delivers the data coming from the multichannel soil characteristics monitoring (JXCT sensor). Additionally, through the one-wire communication port, the ESP32 receives the information regarding the air quality conditions based on the usage of DHT22. Air relative humidity and temperature are measured and delivered to the ESP32, where the information is extracted using DHT MicroPython library.

The soil and air data are delivered, together with the timestamp of the measurement and localization of the smart sensor node, via the Wi-Fi network, to the Cloud Server, which is a virtual private server (VPS) hosted by Digital Ocean. Having in consideration the fact that the autonomy of the smart sensor unit is very important, configuration information, such as localization and Wi-Fi credentials, can be provided from the user’s phone via Bluetooth at the installation stage of the unit.

### 4.2. Embedded Software

The ESP32 embedded software was developed in MicroPython using the Thonny IDE. Several libraries related to communication with UART communication with JXCT integrated soil sensor were used as an important part of developed firmware. Additionally, the DHT library was used to extract the values of air relative humidity and temperature on the considered sites.

Regarding the data communication for this work the Wi-Fi communication protocol was used. Thus, the MicroPython Networking library was used and the WLAN functions. The general flowchart of the main embedded software developed in MicroPython is presented in Figure 6.

Once the controller boots, it first checks if it has any cached configuration, which needs to be provided by the user when installing the sensor. In this configuration stage, the Wi-Fi credentials, the identifier assigned to the micro-controller, and the identifiers and types of the sensors and measurement channels from the cloud and measurement intervals, as well as a web token for secure communication with the server, are passed to the microcontroller. Next, the microcontroller, connects to the Wi-Fi using the passed credentials in the configuration stage and updates its clock using Network Time Protocol (NTP). Finally, the microcontroller starts reading data from the sensors attached to it and sending it to the cloud for centralized storage.

It is very simple to extend the responsibilities of a sensing node, i.e., what parameters it should measure, as the implemented software is made to be modular and scalable. The microcontroller implementation provides to the controller all it needs to interact with various types of sensors, namely, DHT, capacitive, resistive, and TDR. Hence, if we want to add another sensor to the microcontroller, we need only to plug it in to the microcontroller board and reboot the controller to force it to update its configuration.

### 4.3. Mobile App

The mobile app was implemented using the Kotlin Multiplatform Framework (KMM), a framework for building cross-platform applications using the Kotlin language, and serves two purposes, to configure sense nodes over low-power Bluetooth and to visualize system data. As such, the app has two focus areas: interaction with sensors and data analysis. As illustrated in Figure 6, after booting, the sensing node tries to find a configuration if it has been flashed; otherwise, it waits for the user to provide configuration. To provide a configuration to the controller, the user can create one using the mobile application and send it to the node using a low-power Bluetooth communication. A configuration contains identifiers for controller, sensor, and measurement channels, the current location, and WIFI credentials, if necessary, as can be seen in Figure 7.

The data visualization focus area provides data on the latest read quantities and ratio of nutrients in the soil and data on soil moisture, temperature, pH, and salinity. The design of the app was inspired by the app of the Tropical Botanical Garden of Lisbon [105]. It uses a styled map of the garden powered by Mapbox, a cheaper alternative to Google maps, which combines data from several geographic databases, such as OpenStreetMaps [106]. The map was developed considering the inaccuracy of data from Google Map on identifying in that small area the points of interest. On this map, the user can select a specific measurement site to obtain data on soil and air conditions at that specific site or obtain general information on environmental conditions around the garden, as can be seen in Figure 8.

Aside from giving information on the specimens, the user can also obtain directions towards a certain specimen. These features can be helpful for the staff of the garden to easily locate the species and can further include the visitors into the management of plants in the garden.

### 4.4. Experimental Protocol

Laboratory tests were performed to test sensibility of the sensors. A volume of 1000 mL soil was weighted and, after this, was put in the oven for 24 h. After cooling in an open area at room temperature, the NPK sensor was put in the dried sample. Values of temperature and pH were obtained. The other parameters were 0. After this, 10 mL of water was added incrementally to the soil. Data on soil N, P, and K were obtained starting from 5% soil moisture. The experiment was repeated 3 times.

Field tests were performed in Lisbon Botanical Garden. Sensor nodes were deployed at 0.3 m from the trunk of 12 trees of the Tropical Botanic Garden of Lisbon, selected from the 20 available on the tour “Trees You Must See” provided by the Mobile Garden App reported on [105]. Sensor nodes were positioned at 0.3 m from the trunk of each tree, and measurements of soil moisture, temperature, pH, electrical conductivity, and concentrations of N, P, and K, as well as air humidity and temperature, were collected from each site. It should be highlighted that measurements were collected under two conditions, before and after precipitation, with the objective of observing what change the wetness of the soil induced by the rain has on the measured values by the soil sensor. In Table 5, the species whose soil parameters were monitored are listed.

Furthermore, the selection was based on the location, more specifically, the proximity to water bodies and the inclination of the location. The garden covers an area that is slightly inclined; its northmost point is located at 40 m above sea level, while its southmost point is located at only 15 m above sea level. Hence, it was assumed that there would be differences in moisture and nutrient levels at different heights. The garden has also several lakes and brooks, for which, due to evaporation, it was assumed that the moisture would be higher near these bodies of water. In addition to air humidity conditions, plants were selected in relation to their proximity to bodies of water to determine whether this had any influence on the moisture of the soil in the region of these species. An additional criterion was the origin of the species, as they come from different continents with different weather conditions; some come from dry regions, while others are typically found in rainforests. The locations of the sites can be viewed in the Tropical Botanical Garden map (Figure 9) from the Garden’s mobile app.

## 5. Results

Analyzing the measurements of air relative humidity (RH) (Figure 10), it can be observed that higher values were measured near the *Beaucarnea recurvata* (site 1), which is expected, as it is located near the lake, near the *Dracaena draco* (site 5), and *Bauhinia variegata* (site 10). On the other hand, *Afrocarpus mannii* (site 6) and *Phytolacca dioica* (site 11) registered the lower humidity values, as they are farther away from the lake.

Concerning the soil moisture measurement results presented in Figure 11, it can be observed that the highest levels of moisture were verified for *Dracaena draco* (site 5) and *Phytolacca dioica* (site 11). *Brahea edulis* (site 8), *Encephalartos lebomboensis* (site 3), and *Metrosideros excelsa* (site 12) have registered the lowest values. It can also be observed that, in most cases, the values registered after the rain are higher than the values before the rain, with significant growth being registered for sites 1, 11, and 12. In addition, site 5 is the only one where the moisture before precipitation is higher than after.

Regarding soil electric conductivity (Figure 12), an increase can be observed on all sites, which is coherent with the increase in soil moisture. We can observe a larger increase for *Phytolacca Dioica* (site 11) and *Metrosideros excelsa* (site 12), which is also coherent with the measurement results obtained for soil moisture. Moreover, the conductivity is lower before and after the rain for site 5, which is consistent with moisture data.

Soil pH affects various chemical and biological processes. The ideal pH is close to neutral, as most nutrients are available within the 6.0 to 7.5 range of pH, which is very important for plant growth [76,107]. The very acidic soils generally are the result of overapplication of fertilizer and can lead to a decrease in bacteriological activity [108]. The soil microbiome (e.g., cyanobacteria, rhizobacteria, and mycorrhizal fungus) is an important producer of nutrients and organic nutrient regeneration in the soil [109]. However, the preference of some species of plants on slightly acidic soil should be considered. In many measurement sites after rain (Figure 13), the soil pH increases. This condition is favorable for most of the plants, as their metabolism and growth are optimal at a neutral value of pH.

Based on the analysis of the measurements obtained for pH before and after the rain (Figure 13), we can observe for five sites there was a slight decrease in pH after the rain. It should be highlighted that for site 5, for which a decrease in soil moisture was registered, an increase in soil pH was observed. Therefore, in those sites, the influence of other variables than quantity of fertilizers should be considered. Overall, values for pH indicate the soil in the garden is relatively acidic, being possibly associated with higher quantity of fertilizers or soil organic matter.

In Figure 14, the average air temperatures per measurement sites are displayed. Overall values varied from 21 °C to 30 °C. The highest value was observed for site 7, while the lowest value was observed for site 1. The spatial distribution of plants in the garden has great importance when interpreting data on soil moisture content and availability of the nutrients for the plants.

When compared with air temperature, soil temperatures (Figure 15) show less variation and a clear trend can be observed towards 20 °C, which remains stable before and after the rain. In addition, in most cases, there was a slight temperature drop after rain. The highest temperature values were observed for sites 1 and 2 before the rain, and sites 7 and 8 after the rain.

Nutrient concentrations of N, P, and K are proportional, and this can be observed in the graphs of Figure 16. It can be observed that K is found in larger quantities than P and N, and the concentration of N, P, and K has increased significantly after rain in most cases, especially for site 11 (*Phytolacca dioica*) and site 12 (*Metrosideros excelsa*). Information on soil texture and soil organic matter should be considered in the interpretation of the changes in NPK sensor data after rain, as those variables are associated with the distribution of macronutrients with increase in soil moisture. The increase in the measured concentrations for N, P, and K is coherent with the increase in electric conductivity. It could be observed that, for most cases, there was an increase after the rain and, only in sites 5 and 8, the value of acquired data on soil N, P, and K decreases. The same pattern could be observed for electric conductivity.

The need for data on other soil parameters (e.g., soil texture, foliage cover, slope of landscape, and drainage) is underlined by data from the correlation matrix of measured soil parameters before and after precipitation (Figure 17 and Figure 18). If, before precipitation, there was a strong correlation of data on soil moisture, electric conductivity, and concentrations of N, P, and K, after precipitation, this correlation slightly weakens.

## 6. Discussion

Information on research realized for developing an IoT-based information system for soil nutrients characterization for small-sized landscape with potential application on horticulture is presented. The developed system may help in assessment of soil nutrient changes for better management of irrigation and fertilizer administration. The system has capacity to acquire data on soil N, P, and K content, soil moisture, temperature, conductivity, and pH, as well as data on air temperature and humidity. IoT protocols were considered for device connectivity and services. The developed system includes a mobile app that enables visualization of sites/datapoints where data were collected, as well as graphical visualization of changes in the acquired data. The system prototype enables assessment of soil nutrient changes, as well as changes in soil water content (moisture data) and changes in soil pH. For comprehensive interpretation of data from NPK sensors and detection of potential errors in sensor data acquisition, the system also includes data on conductivity, soil moisture, temperature, and pH, as well as air temperature and humidity.

Tests were realized in the laboratory and in Lisbon Botanical Garden. The majority of the requirements for an IoT-based information system that may be used in this garden for soil nutrients assessment are similar with those for a horticulture farm that may include pomiculture, olericulture, or floriculture (e.g., digital maps with better spatial resolution than on Google map; low cost but reliable, portable sensor for soil nutrient quantification; sensing data on environmental factors that may support interpretation of soil nutrient sensor; affordable connectivity of devices; easy to use; and support for easy maintenance and upgrade).

The system demonstrated good performance in the field experiment, as in places located near bodies of water, such as lakes and brooks, higher values of air humidity and an overall increase in moisture and conductivity after precipitation were registered. An increase in air humidity at the measurement sites was also identified. As the soil sensor from the sensing nodes is based on time delay reflectometry, it is affected by soil dielectric constant and an increase in the water content in the soil resulted in an increase in the values for all channels that use this principle besides moisture, namely electric conductivity and concentration of macronutrients. It was also observed that, although surface air temperature varied between 21 and 30 ºC, soil temperature variation was less pronounced and remained stable throughout the measurement sites, although a slight decrease was observed in most places after precipitation events. For comprehensive interpretation of a slight increase in soil temperature in sites 7 and 8, data on soil texture and soil foliage should be included, as, in some places, enhanced activity of microbiome after rain increases soil temperature. Furthermore, the soil at the measurement sites showed to be slightly acidic. An inverse correlation was observed between soil pH and moisture.

The collected data underscore the necessity of including in soil nutrient assessment, using the developed information system, the data on soil texture, with special detail on the superficial layer (i.e., foliage cover), as this layer may influence soil permeability and activity of the microbiome. The data on texture may be included by a user of a mobile app after qualitative measurement by visual observation and hand test. For larger areas, public data on soil texture may be used for assessing soil texture in a mobile app. For instance, Land Use Cover Area frame Surveys (LUCAS) topsoil database may be used to extract data on soil texture for an area. The LUCAS database is a European database that was made available in 2016 that includes datasets on soil physical properties (silt, clay, sand, and coarse fragments) for the European Union, together with maps of derived products (bulk density and available water capacity) [110]. Our research, as well as data from the literature, indicates that exact determination of different soil nutrients’ availability for plants requires information on many environmental variables. Optimal information on flow of nutrients in soil may be obtained only by considering the dynamic, stochastic, complex process of nutrient transformation in soil.

Future works are necessary to improve accuracy of estimation of soil N, P, and K by using multi-channel sensing. Moreover, sensing devices that enable in situ sensing of other macronutrients (e.g., calcium) or micronutrients (e.g., iron) should be integrated in an IoT-based information system. In addition to their importance for plant physiology, these elements may influence availability of N or K for plant roots and, as such, greatly influence soil fertility and crop productivity.

Future developments will include the integration of soil texture for fertilizer administration and a customizable application for management of irrigation and fertilizer administration. Maps will also be conceived for the representation of physical properties of soil, the slope, and drainage pattern of landscape considering the topography of some horticulture farms. The affordability of the developed solution should be considered, as well as the adoptability. Farmers’ knowledge and perception on new technology, and incentive policy may have an important role on adoption of an information system for soil nutrient assessment.

## 7. Conclusions

The quality and quantity of plant growth depends mainly on the availability of soil nutrients for the plant. Traditional laboratory methods provide accurate measurements of soil nutrients but are expensive, complex, and time-consuming. Advances in information and communication technology enabled the development of electronic devices that may perform faster measurements of various soil parameters. We present an overview of new techniques and technologies for soil fertility assessment. We described the challenges and work that was done for development of an information system based on IoT for soil nutrient estimation. A wireless smart sensor architecture is proposed that is characterized by multimodal data acquisition capabilities. The system includes data storage and data display supported by cloud and mobile computing. A mobile application allows configuration of the system, visualization of garden map and places of stations for in situ monitoring of the soil and air parameters, and visualization of graphical data. Soil characteristics, such as moisture, pH, electrical conductivity, and temperature, are measured together with the concentrations of macronutrients using a multi-channel time delay reflectometry sensor. The soil data are integrated with data on air temperature and relative humidity. The developed system was tested by performing measurements at different sites and different periods in the Tropical Botanical Garden of Lisbon considering a variety of soils and plants in a relatively reduced area. The analyzed experimental data underline the capacity of the system to provide relevant data on soil parameter changes. The need for multi-channel acquisition of environmental data for better interpretation of data on NPK sensor was underscored by various results on system tests. Future developments of the system may include data on soil texture and data on fertilizer administration. The system may provide important data on flow of soil nutrients in horticultural farms. Information provided by the system may contribute to better management of irrigation and fertilizer administration in horticulture farms.

## Figures and Tables

**Figure 1 sensors-23-00403-f001:**
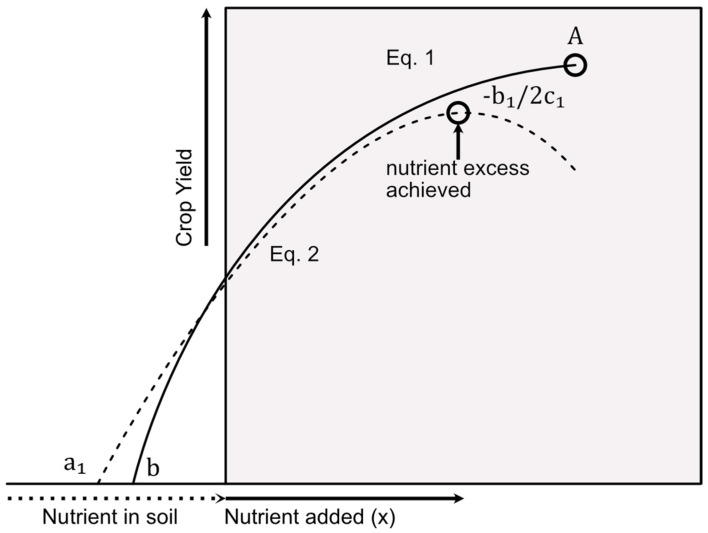
Crop response curves for Equation (1), nutrients present in adequate amount, and 2, nutrients present in excess (based on diagram from [14]).

**Figure 2 sensors-23-00403-f002:**
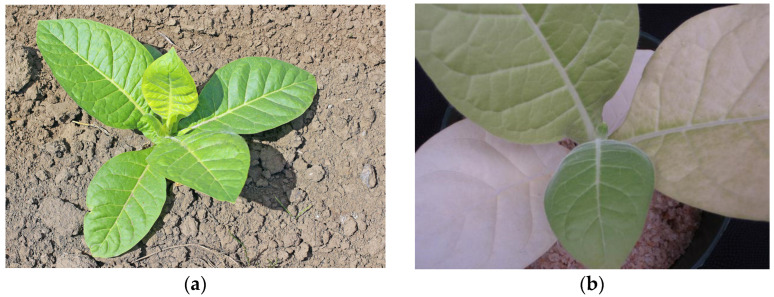
Tobacco plant: (**a**) Healthy tobacco sprout without nitrogen (N) deficiency; (**b**) tobacco with N deficiency [17].

**Figure 3 sensors-23-00403-f003:**
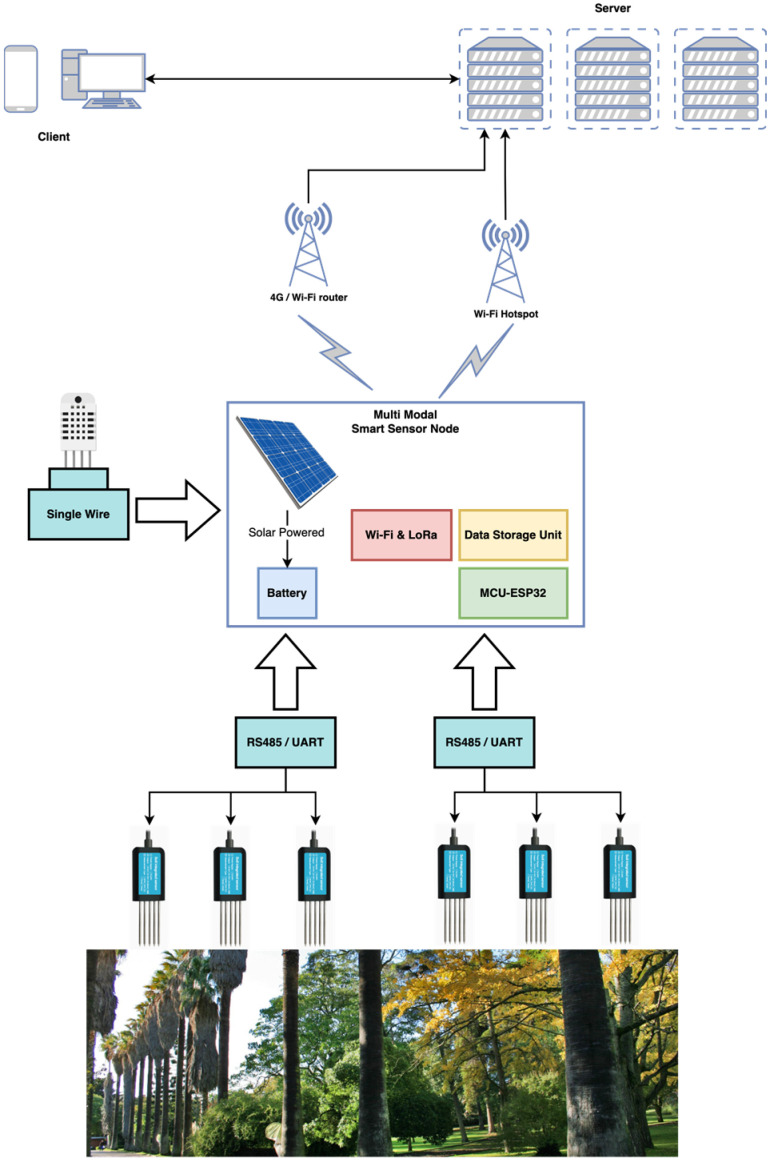
IoT ecosystem architecture for plant stress assessment for Tropical Botanical Garden of Lisbon.

**Figure 4 sensors-23-00403-f004:**
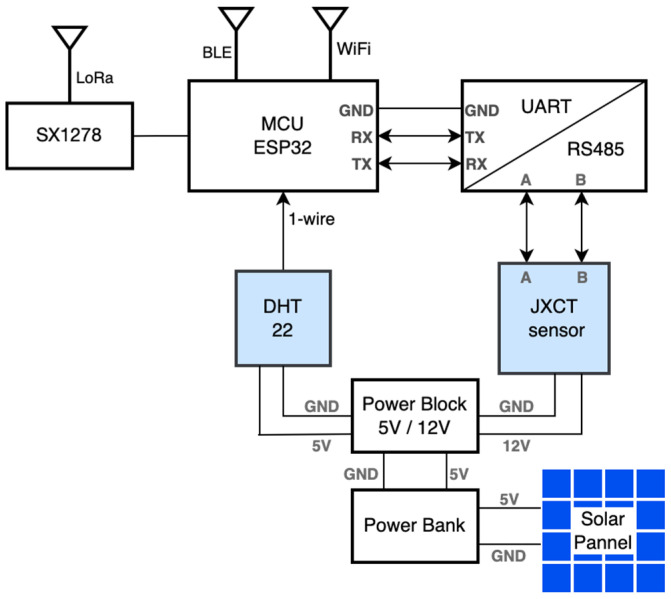
Soil smart sensing node block diagram.

**Figure 5 sensors-23-00403-f005:**
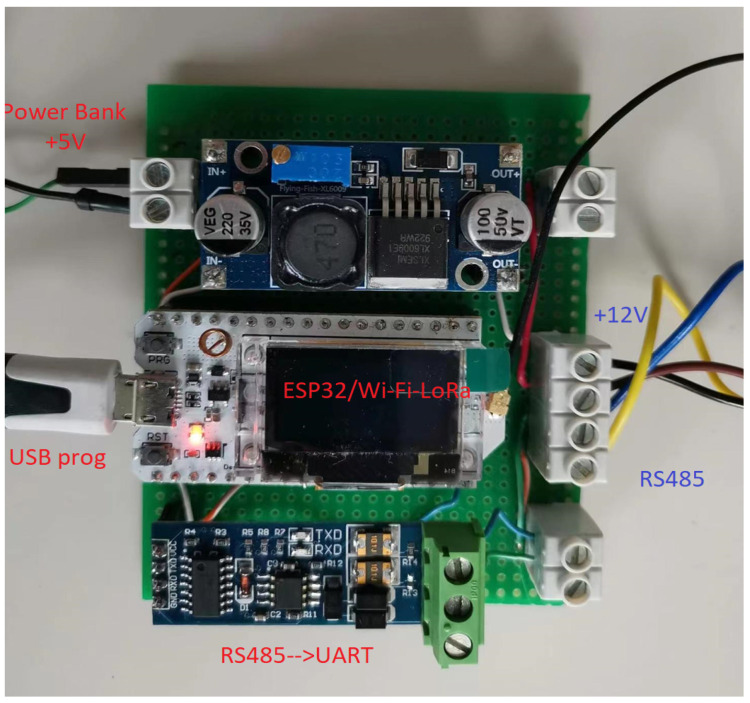
Smart sensor node implemented prototype.

**Figure 6 sensors-23-00403-f006:**
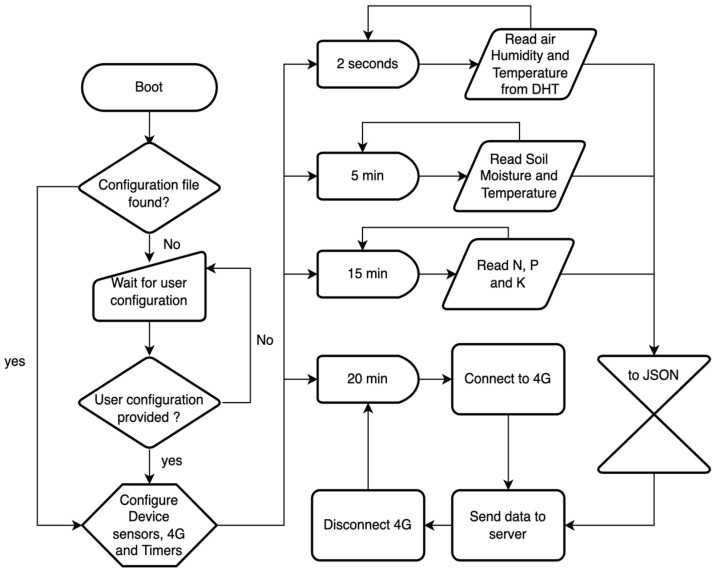
Flowchart of microcontroller embedded software.

**Figure 7 sensors-23-00403-f007:**
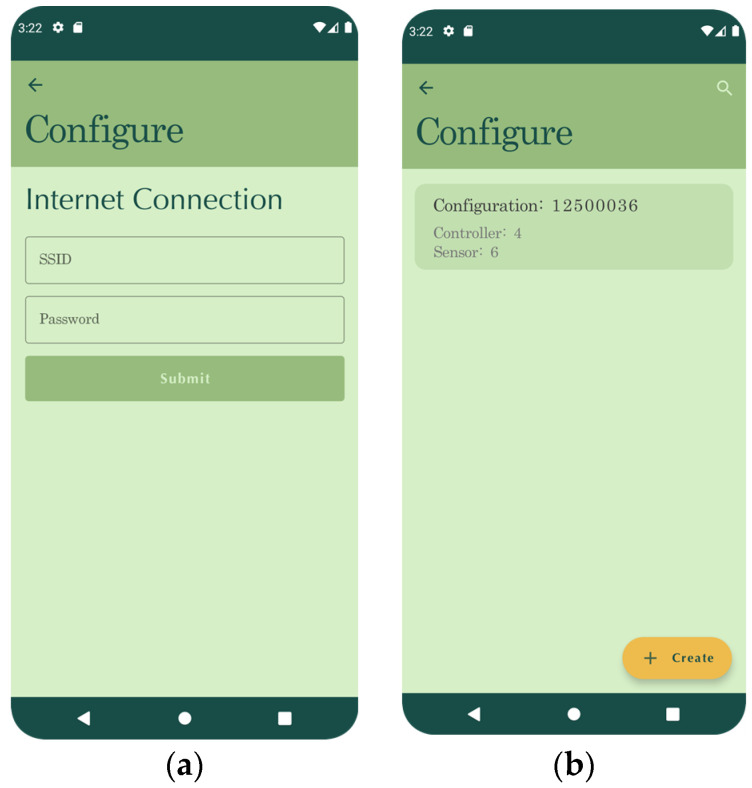
Configuration screens: (**a**) Wi-Fi configuration screen; (**b**) sensing node configuration screen.

**Figure 8 sensors-23-00403-f008:**
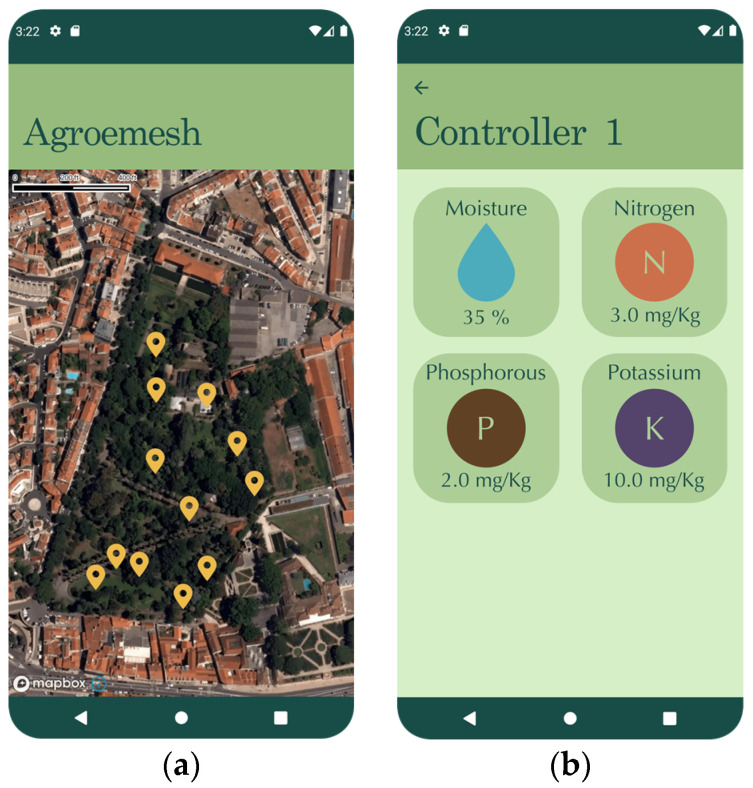
Agroemesh and Controller screens: (**a**) Agroemesh Map screen; (**b**) Soil status screen for selected controller localization.

**Figure 9 sensors-23-00403-f009:**
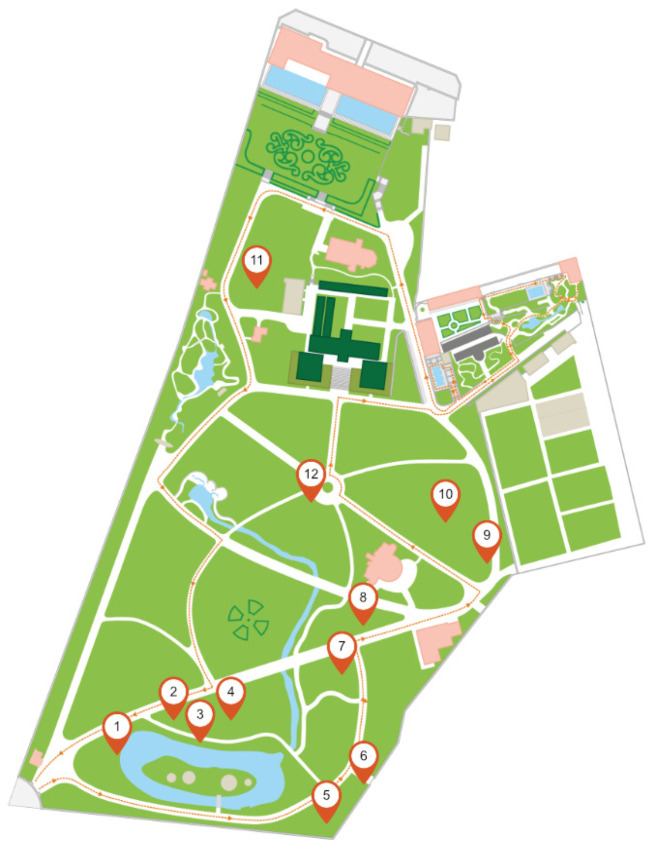
Map measurement sites where the numbers from 1 to 11 represents the labels for the measurement sites.

**Figure 10 sensors-23-00403-f010:**
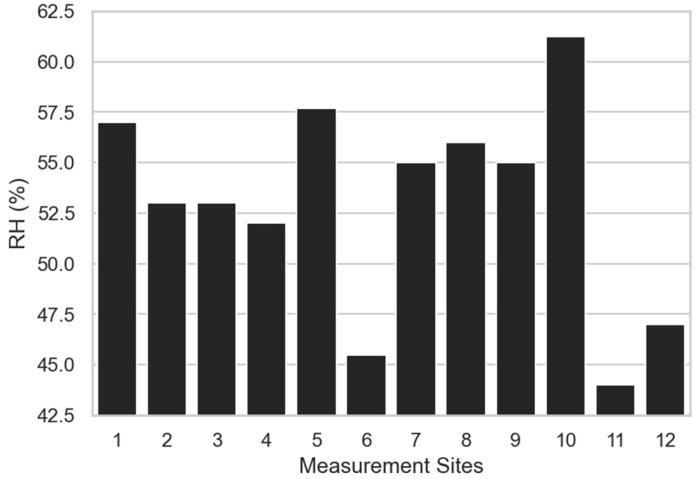
Air relative humidity at the measurement sites.

**Figure 11 sensors-23-00403-f011:**
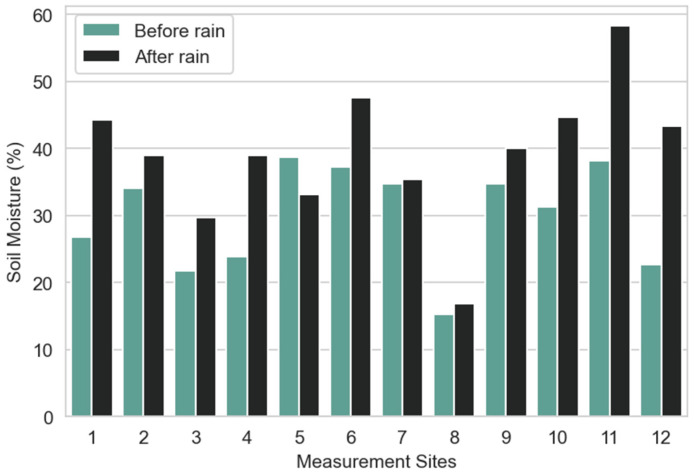
Average soil moisture per measurement site before and after rain.

**Figure 12 sensors-23-00403-f012:**
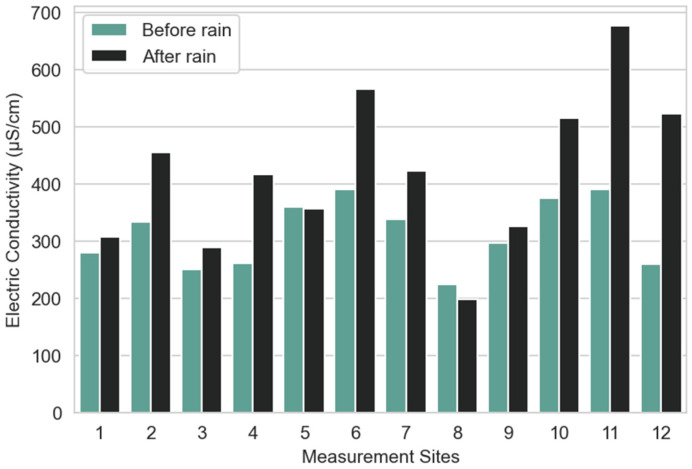
Average soil conductivity per measurement site before and after rain.

**Figure 13 sensors-23-00403-f013:**
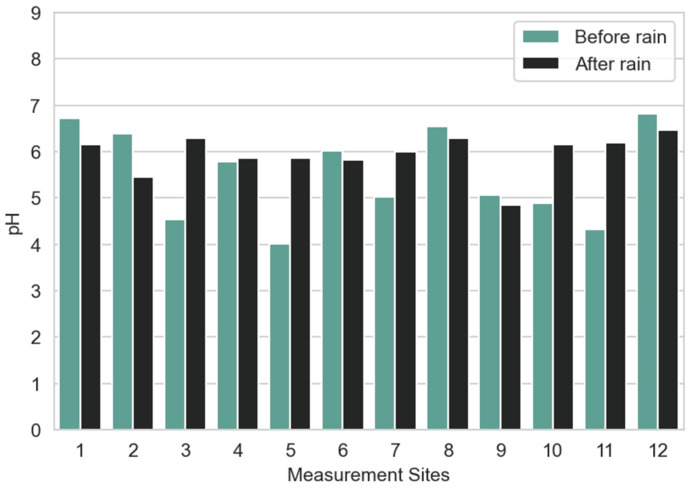
Average pH per measurement site before and after rain.

**Figure 14 sensors-23-00403-f014:**
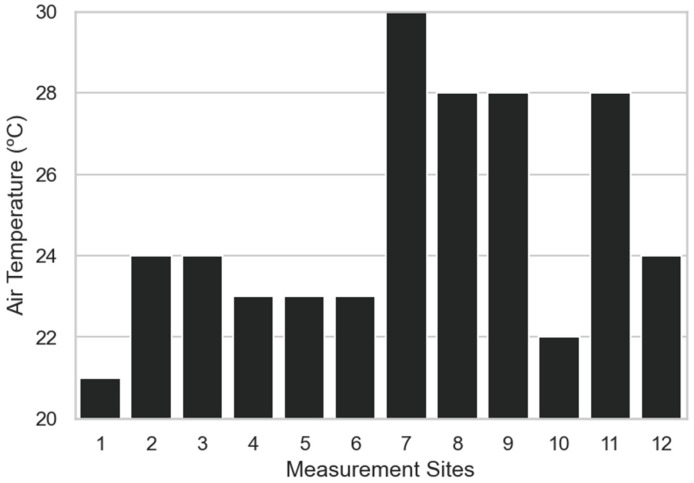
Average air temperature per measurement site.

**Figure 15 sensors-23-00403-f015:**
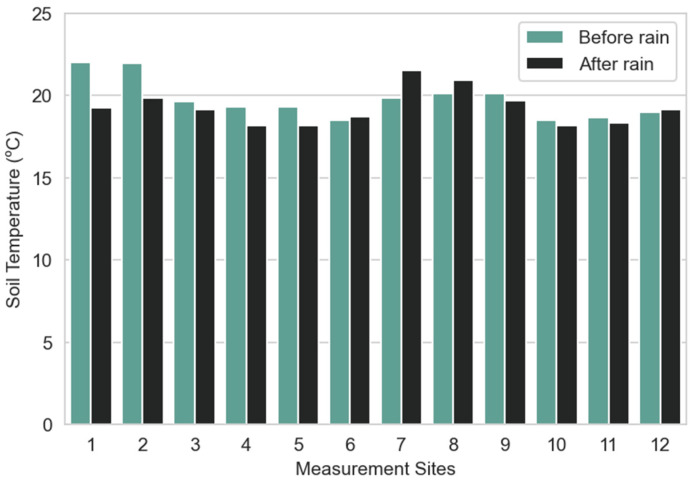
Average soil temperatures per measurement site before and after rain.

**Figure 16 sensors-23-00403-f016:**
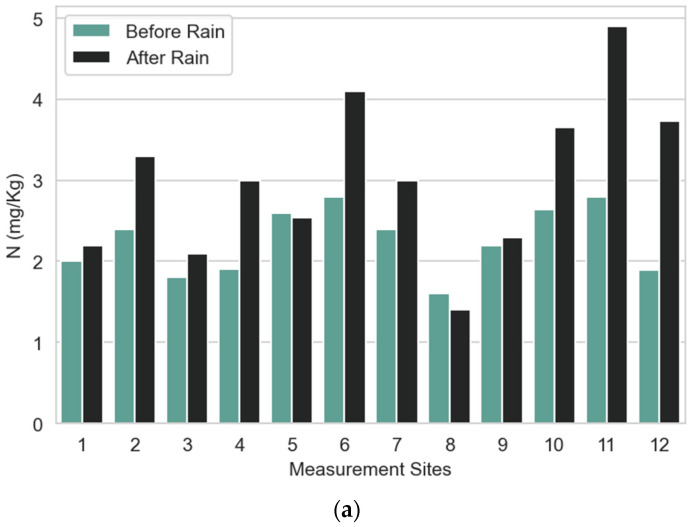
Average soil N, P, and K data before and after rain: (**a**) N data; (**b**) P data; and (**c**) K data.

**Figure 17 sensors-23-00403-f017:**
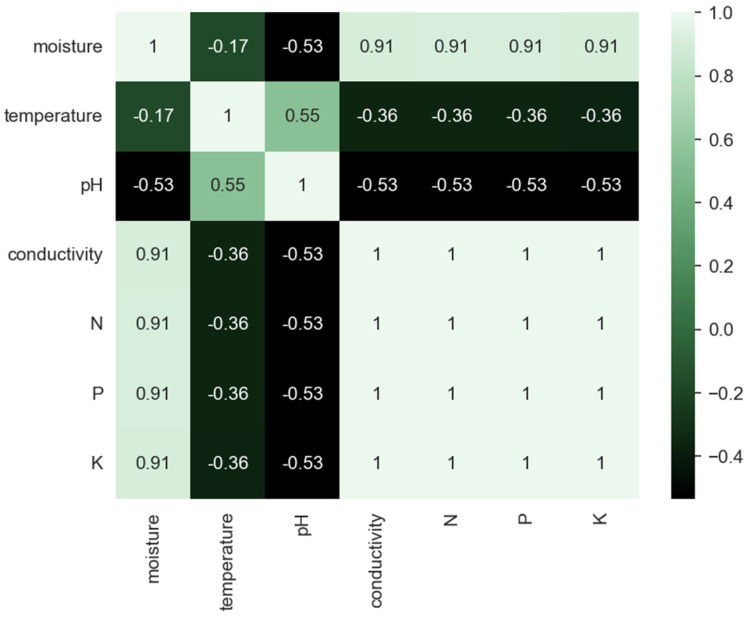
Correlational matrix for soil parameters from values collected before precipitation (where the cell colors correspond to the scale presented on the right side where negative values correspond to dark green and positive values to light green).

**Figure 18 sensors-23-00403-f018:**
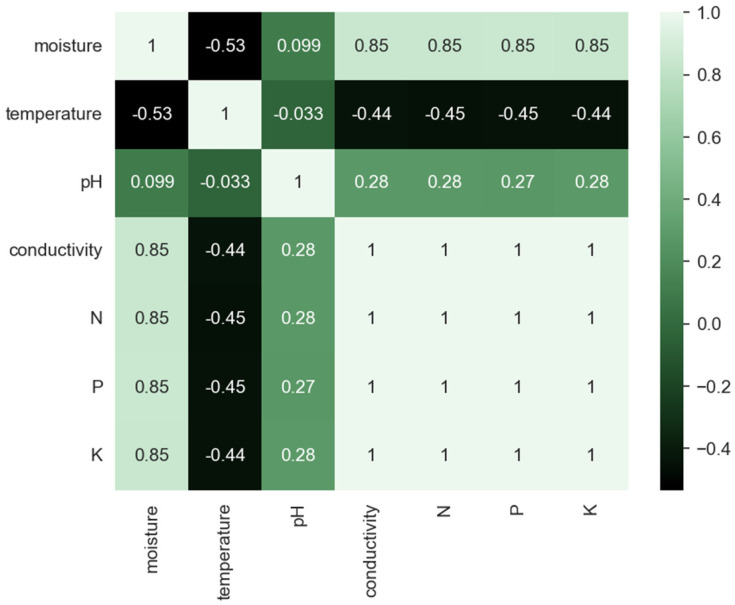
Correlational matrix for soil parameters from values collected after precipitation (where the cell colors correspond to the scale presented on the right side where negative values correspond to dark green and positive values to light green).

**Table 1 sensors-23-00403-t001:** Methods of soil fertility assessment, advantages, and disadvantages.

Method	Type	Advantages	Disadvantages
Biological characterization	Qualitative	Easy to obtain data; may provide data on influence of external factors and spatial variability	May be affected by differences on evaluator perception and knowledge
Plant analysis	Qualitative or semi-quantitative	Fast, easy to obtain data	Changes are observed when it is too late.
			Time consuming, some procedures are expensive
Chemical and physical characterization	Quantitative	Precise, efficient	Some portable sensors have lower sensibility, sensitivity, and accuracy than laboratory assessment

**Table 2 sensors-23-00403-t002:** Nutrients for plant growth (adapted from [22]).

Essential Plant Elements			Beneficial Plant Elements
** *Organic Elements* **	** *Mineral Elements* **		Cobalt (Cb)
Carbon (C)	**Macronutrients**	**Micronutrients**	Silicon (Si)
Hydrogen (H)	**Primary**	Boron (B)	Sodium (Na)
Oxygen (O)	Nitrogen (N)	Chlorine (Cl)	
	Phosphorous (P)	Iron (Fe)	
	Potassium (K)	Manganese (Mn)	
	**Secondary**	Molybdenum (Mo)	
	Calcium (Ca)	Nickel (Ni)	
	Magnesium (Mg)	Zinc (Zn)	
	Sulfur (S)		

**Table 3 sensors-23-00403-t003:** Average concentrations of mineral elements in plant shoot dry matter sufficient for adequate growth [23].

Element	Chemical Symbol	Concentration (μmol g^−1^ DW)	Concentration (mg kg^−1^)
Nitrogen	N	1.000	15,000
Potassium	K	250	10,000
Calcium	Ca	125	5000
Magnesium	Mg	80	2000
Phosphorus	P	60	2000
Sulphur	S	30	1000
Chlorine	Cl	3.0	100
Boron	B	2.0	20
Iron	Fe	2.0	100
Manganese	Mn	1.0	50
Zinc	Zn	0.3	20
Copper	Cu	0.1	6
Nickel	Ni	0.001	0.1
Molybdenium	Mo	0.001	0.1

**Table 4 sensors-23-00403-t004:** Time delay reflectometry integrated soil sensor characteristics.

Soil Parameter	Measurements Range	Measurements Accuracy
Temperature	−40 °C–80 °C	±0.4% of FS
Electric Conductivity	0–20 mS cm^−1^	±2% of FS
Moisture content	0–100%	±2% of FS (0–50%) ±3% of FS (50–100%)
pH	3–9	±5% of FS
N content	1–1999 mg/Kg	±2% of FS
P content	1–1999 mg/Kg	±2% of FS
K content	1–1999 mg/Kg	±2% of FS

FS = full scale.

**Table 5 sensors-23-00403-t005:** Measurement sites and species.

Measurement Site	Species
1	*Beaucarnea recurvata* Lem.
2	*Sequoia sempervirens* (D. Don) Endl.
3	*Encephalartos lebomboensis* Verd.
4	*Ficus macrophylla* Pers.
5	*Dracaena draco* (L.) L.
6	*Afrocarpus mannii* (Hook.f.) C.N.Page
7	*Araucaria bidwillii* Hook.
8	*Brahea edulis* H.Wendl. ex S.Watson
9	*Ceiba speciosa* (A.St.-Hil.) Ravenna
10	*Bauhinia variegata* L.
11	*Phytolacca dioica* L.
12	*Metrosideros excelsa* Sol. ex Gaertn.

## Data Availability

Not applicable.

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
