# Peer review of "IoT-Based Systems for Soil Nutrients Assessment in Horticulture"

_sensors, 2022, doi:10.3390/s23010403_

Round 1
Reviewer 1 Report
Dear authors,
Thank you for the opportunity to review this Manuscript (State of fertility and soil management methods: review and Botanical Garden case study). This article makes a review of existing methods for monitoring of soil conditions, with focus on soil nutrients and factors that affect uptake of these nutrients by the plant, namely soil moisture, temperature, pH, salinity and air humidity and temperature. In addition, a system for monitoring soil parameters for various species around the garden is presented, which was used in a field experiment in the Tropical Botanical Garden of Lisbon.”
There is some aspect that should be reviewed by authors.
The manuscript is not clear if it is a paper or a review.
In the abstract, there is no information about the study with the objective, Results and discussion. Based on this observation, the abstract should be rewritten.
In the abstract: Add this “This article makes a review of existing methods for monitoring of soil conditions, with focus on soil nutrients and factors that affect uptake of these nutrients by the plant, namely soil moisture, temperature, pH, salinity and air humidity and temperature. In addition, a system for monitoring soil parameters for various species around the garden is presented, which was used in a field experiment in the Tropical Botanical Garden of Lisbon.”
The topic “2. State of the art” did not bring innovation for the study.
The topic 3. It looks like Material and Methods. However, It is not clear.
This topic “Results and Discussion” should focus on Results. And I recommend the Discussion as other topic.
How many years of experimental?
Information of study are.
The Discussion is poor
Reviewer 2 Report
Dear Editor,
Subject: Review for Sensors
General comments
The present study, "State of fertility and soil management methods: review and Bo- 2
tanical Garden case study" manuscript number # sensors-2080154, conducted by Postolache et al., assesses the state of soil fertility and soil management which is one of the hot topics in recent agriculture to increase crop production, and decrease soil degradation and environmental pollution. I consider the topic is still relevant, as quantifying and understanding the effect of organic and inorganic fertilization on soil fertility and soil management on sustainable basis. As the authors reported, the application of chemical fertilizer in low amount or in combination with organic fertilization is a good strategy to increase soil fertility and crop growth while decreasing soil degradation.
According to the manuscript title, I suggest the authors need to write one to two sentences in the abstract section on soil management. Overall the manuscript is well written and well explained.
I have some comments that should be taken into account when revising this manuscript:
Comments and Suggestions for Authors:
Title. Please try to make the title clear, restructure the “review and botanical garden case study”.
Line 14 and 24. The author need to be consistence, in some case the author used “fertiliser” and in some case “fertilizer”, need to check the entire manuscript.
Line 32. The first sentence of the introduction section needs to be revised.
Line 31_71. The introduction having repetitions and have no connection, please improve the introduction. Once more the authors only focused on Botanical Garden in the introduction section, there is no single sentence regarding soil fertility nor management? Please consider these suggestions.
Line 76-79. The sentence is too long and not clear, please divide it in two or three sentences to make it readable.
Line 77. The repeated word “soil” should be deleted from “soil nutrients and soil fertility” or replace with “soil nutrients and fertility”.
Line 80-82. Please add a reference at the end of the sentence “Therefore, for optimal production and to preserve the fertility of the soil it is necessary to maintain adequate amounts of all necessary nutrients”
Line 83-86. Please to be consistence, each word of “Biological Assessment, Plant Analysis, Soil Testing” are capitalized but not for “Spectral Sensing and Site-specific fertility evaluation”.
Line 91 and 99. These equations need to be referenced.
Line 157. Soil Organic Matter should be soil organic matter.
Lines 165-174. The authors need to use the abbreviation of these nutrients and develop the abbreviation for the first time. I suggest that the authors need to explain these abbreviations in the abstract section as the have written full name of these nutrients.
Lines 557 and 580. Please crosscheck your all citation and reference, and correct these references “Error! Reference source not found”. Please update your citation with the latest one doi: 10.1007/s11356-021-15579-7. Epub 2021 Jul 23.
Conclusions; This section needs to be improved
Reviewer 3 Report
The article ‘State of fertility and soil management methods: review and Botanical Garden case study’ by Stefan Postolache, Pedro Sebastião, Vitor Viegas, Octavian Postolache and Francisco Cercas is an very interesting manuscript. The authors put a lot of work in research and preparing of manuscript. The work describes methods used to determine soil fertility and nutrients, and system for monitoring and analysing nutrient replenishment. This article fits the subject of the Sensors Journal. The manuscript may be published after a slight correction. Some minor flaws are listed below:
Line 141 – where is the explanation of “GIS” abrreviation?
Table 2 – the name of columns in the Table 2 should be given (not only units).
Table 3 – abbreviation ‘FS’ should be explained. Why accuracy of every parametrs is given in % but temperature in ºC?
Line 567 – title of subsection is given double (in line 567 and 587)
Reviewer 4 Report
Please consider the following suggestions that could help to improve the paper:
1. Maybe Figs. 12 (a) and (b) could be better compared using a single histogram. For instance, by columns 2-D or 3-D in Excel software.
2. The same for Figs 13, 14 and 16.
3. Related to Fig. 17, I would suggest using a Correlation Matrix, for instance using R software. It is good for obtaining the dependencies among measurement sites. The same dependencies can be observed using a graph using Principal Component Analysis (PCA).
Minor issues:
4. Fig. 11: Please check line 497 on page 15. Site 11 has the lowest RH.
5. Page 7, line 285. There is a blank.
6. Page 9, Line 353. What does “IoT compatible” mean?
7. Figure 5: Please check the direction of the blue arrows.
8. Page 10, line 386: What does “a1” mean?
9. Table 3. Please use ºC. Also, check the symbol between 0 and 50%
10. Fig. 6. The first letter in “Node” and “Implemented” is better in lowercase letters.
Round 2
Reviewer 1 Report
The manuscript can be published.